# Topology-dependent self-structure mediation and efficient energy conversion in heat-flux-driven rotors of cholesteric droplets

Jun Yoshioka[1] & Fumito Araoka [1]

When heat flux is applied to a chiral liquid crystal, unidirectional rotation is induced around the flux axis, as first discovered by Otto Lehmann in 1900. In recent years, this heat-flux-induced phenomenon has been studied mostly in droplets of cholesteric liquid crystals undergoing phase transition from the isotropic to cholesteric phase, i.e., in the coexistence region, which occurs over a very narrow temperature range. Here, we report that the heat-flux-induced rotation can be stabilised by the use of a dispersion system, in which the cholesteric droplets are dispersed in a viscous and poorly miscible isotropic solvent. Interestingly, the phenomenon is found to be topology dependent. Moreover, the rotation is not only stable but also more efficient than that in the known systems. We describe in detail how the dynamics of the heat-flux-induced rotation are altered in the present dispersion system.

[1] RIKEN Center for Emergent Matter Science (CEMS), 2-1 Hirosawa, Wako, Saitama 351-0198, Japan. Correspondence and requests for materials should be addressed to F.A. (email: fumito.araoka@riken.jp)

At the end of the 19th century, Otto Lehmann first found heat-flux-induced rotation (HIR) of the director field in a liquid crystal (LC) material subjected to a temperature gradient. Currently, this phenomenon is known as the so-called Lehmann rotation[1,2]. A theoretical discussion of Lehmann rotation was provided by Leslie nearly 70 years after Lehmann's observation. He considered constitutive equations of heat flux and explained a possible direct coupling between the heat flux and the director rotation in cholesteric (Ch) LCs[2,3]. In this theory, the rotation is attributed to the existence of chirality, and the induced rotational speed is simply proportional to the strength of the chirality[2]. In spite of this theoretical explanation, in fact, Lehmann rotation had hardly been experimentally reproduced for more than one century after Lehmann's discovery, which suggests that the induced rotational torque could be too weak to be observed in typical Ch LCs. However, a similar HIR phenomenon

in a bulk Ch LC was recently observed by Oswald et al. in 2007[4]. The important finding of this work was that they used a unique sample, a 'compensated cholesteric liquid crystal', in which the Ch helix inversion occurs at a certain temperature, and they succeeded in inducing the rotation even at the inversion temperature at which the macroscopic helix vanished. This result implies that the mechanism of HIR in Ch LCs is not as simple as that predicted by Leslie's theory. At the very least, the mechanism depends strongly on the selection of the component LC materials.

More recently, Oswald et al.[5,6] also reported HIR in Ch droplets formed through metastability in the coexistence region during the phase transition from the isotropic (Iso) to Ch phase. In this case, each droplet shows clear monodirectional rotation about the gradient axis, and the conversion from the heat flux to the rotation appears to be much more efficient than in bulk; i.e., the angular velocity of HIR becomes larger in droplets than in

| | Type-A | Type-B | Type-C | Type-D | Type-E1 | Type-E2 | Type-E3 |
|---|---|---|---|---|---|---|---|
| | **a** | **b** | **c** | **d** | **e** | **f** | **g** |
| Experimental POM | | | | | | | |
| | **h** | **i** | **j** | **k** | **l** | **m** | **n** |
| Simulated POM | | | | | | | |
| | **o** | **p** | **q** | **r** | **s** | **t** | **u** |
| Schematic side-viewed cross-section of director field at mid-plane | | | | | | | |
| Range of $N = 4R/P_0$ | $N < {\sim}2$ | ${\sim}2 < N < {\sim}6$ | ${\sim}6 < N < {\sim}9$ | ${\sim}6 < N < {\sim}9$ | ${\sim}9 < N$ | ${\sim}9 < N$ | ${\sim}9 < N$ |
| Number and type of defect | One central point defect | One central point defect | One surface point defect | One central- and two surface point defects | One surface coiled defect | One surface coiled defect | One surface coiled defect |
| Symmetry when confined in spherical-cap boundary | $C_\infty$ ∞-fold axis // Surface normal | $C_\infty$ ∞-fold axis // Surface normal | $C_1$ | $C_2$ Two-fold axis // Surface normal | $C_2$ Two-fold axis // Surface normal | $C_1$ | $C_1$ |
| Orientation of helix when confined in spherical-cap boundary | No helix | // Surface normal | // Surface normal | // Surface normal | ⊥ Surface normal (short pitch) // Surface normal (long pitch) | Tilted | ⊥ Surface normal (long pitch) // Surface normal (short pitch) |
| Rotation conversion efficency $\eta$ (rad mm$^2$ mJ$^{-1}$) | No rotation | No rotation | 0.6~0.9 (High) | 0.2~0.4 (Moderate) | 0.05~0.08 (Low) | 0.05~0.13 (Low) | 0.1~1.0 (High) |

**Fig. 1** Summary of experimentally obtained POMs, simulated POMs and topological features of dispersed Ch droplets. **a–g** Polarising micrographs of five types of Ch droplets dispersed in a fluorinated oligomer: **a** Type-A, **b** Type-B, **c** Type-C, **d** Type-D and **e–g** Type-E droplets. Type-E droplets are further sorted into three sub-types, namely, **e** Type-E1, **f** Type-E2 and **g** Type-E3. Scale bars, 10 μm. The concentrations of the chiral dopant are 0.2 wt% in **a**, 0.5 wt% in **b**, 1.0 wt% in **c**, **d** and **g**, and 2.0 wt% in **e** and **f**. **h–n** Simulated POM images by means of the Jones matrix method, corresponding, respectively, to **a–g**. The structural formulations for these structures are described in Supplementary Notes 1–4 (Supplementary Equations (1)–(8)). **h** Type-A and **i** Type-B are calculated based on the director distributions given by Supplementary Equations (1) and (2). **j** Type-C is calculated with Supplementary Equations (3) and (4) and (**k**) Type-D with Supplementary Equations (5) and (6). **l** Type-E1, **m** Type-E2 and **n** Type-E3 are calculated with Supplementary Equations (7) and (8) with three different helix directions **h** with respect to the cell normal (=z-axis//temperature gradient), i.e., **l**, **n** and **m** correspond to h⊥z, h//z, and h inclined from the z-axis, respectively, (see Fig. 9c). For these droplet types, the structural properties and physical features of HIR, i.e., the conditions of the factor N, number and topological types of defects, symmetry groups, orientations of the helical axis, and η values, are summarised together. r. (**o–u**) 3D schematics of the calculated director fields at the midplane (see also Figs 8 and 9 for other cross-sections)

bulk even under the same heat flux conditions. There are two ways of explaining this experimental fact. First, the director field is mobile, sliding along the boundary, because of the existence of the isotropic slippery interface in the dispersion system[7–9]. Second, the physical mechanism of HIR in the Ch droplets is intrinsically different from that in bulk[5,6,10,11]. Another notable example of HIR in LC droplets was found in an achiral lyotropic nematic (N) LC;[12] i.e., a similar monodirectional rotation about the gradient axis was induced in the metastable droplets formed in the Iso-N transition. In fact, the inner N director field of each droplet was twisted even in this case, and in spite of the achiral molecules, the droplet could be regarded as an analogue of the Ch droplet. Thus, the rotational direction was clearly dependent on the sense of the director twist. Based on these observations, it is natural to think that the twisted arrangement is at least one of the requirements for HIR in LC droplets.

To summarise, HIR in LC droplets has thus far been demonstrated mostly in transient states of narrow temperature regions of several degrees between the Iso and Ch (or lyotropic N) phases of systems composed of mixtures of LC materials[5,6,10–13]. Therefore, precise control of the temperature is required for the appearance of HIR. To realise the steady rotation in a wider temperature range, especially in the regions near room temperature, it is strongly desirable to stabilise the Ch droplets in dispersions or emulsions, for example, by mixing them with an immiscible medium such as water, glycerol or dimethylpolysiloxane (DMPS)[14–18]. However, according to our preliminary studies, no HIR could be observed in such dispersed states. This finding means that the rotational torque could be too weak or even non-existent in these dispersed states, and hence, obviously HIR is governed not only by the twisted N directors but also by the substance surrounding them. Thus, to date, there has been no successful report of HIR in a dispersion state or in an emulsion state.

In this work, we demonstrate a successful realisation of HIR in a dispersion system in which Ch droplets are immersed in an isotropic oligomeric fluid that is slightly miscible to the LC. This finding indicates that for the realisation of HIR, the entire system does not need to be composed of a single LC material, as in the previous reports. Interestingly, the HIR in the present dispersion system is topology dependent and is more stable and efficient than that in the previously reported systems composed of single LC materials; i.e., both the observable temperature range and rotational speed are larger by an order of magnitude. These properties are vital in further investigating the origin of HIR as well as for the potential applications in energy conversion, such as rotational generators.

## Results

**Topological diagram of cholesteric droplets in the present dispersion system.** The topological state of the droplets surrounded by homeotropic surfaces varies depending on the size of the droplets and the chirality strength[18–23]. As distinguished by their appearance under polarising microscopy (POM) (Fig. 1a–g), we classify the topological states of our Ch droplets dispersed in a slightly miscible fluorinated oligomeric solvent (PF656, Omnova Solutions Inc.) into five different types: Type-A (cross pattern, Fig. 1a), Type-B (twisted cross pattern, Fig. 1b), Type-C (U-shaped pattern, Fig. 1c), Type-D (figure-eight pattern, Fig. 1d) and Type-E (tumbleweed-like streak pattern, Fig. 1e–g). Type-E can be further subdivided into three groups, also based on the appearance under POM: Type-E1 (long-pitch streaks), Type-E2 (short-pitch streaks) and Type-E3 (concentric streaks), as shown in Fig. 1e–g. These three Type-E groups are considered to be almost identical in structure to one another, as will be discussed

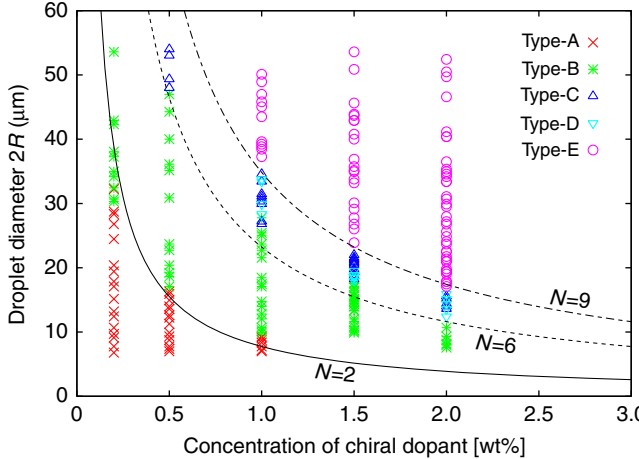

**Fig. 2** Phase diagram representing the stability of Type-A through Type-E. The stationary temperature is 60 °C. The three curves are drawn with the function $N = 4R/P_0$ for $N = 2$ (solid line), $N = 6$ (broken line), and $N = 9$ (dashed dotted line), which can be approximated to be the boundaries between Type-A and Type-B, Type-B and Type-C/D, and Type-C/D and Type-E, respectively

later. The stabilities of these five (Types-A–E) states can be differentiated only by one parameter, $N = 4R/P_0$, which is the ratio of the half helical pitch $P_0/2$ and the droplet diameter $2R$[19–23]. $P_0$ is the pitch of the Ch helix as observed in an LC cell, which corresponds to the strength of chirality. In the present system, the droplets are found to be Type-A for $N < \sim 2$, Type-B for $\sim 2 < N < \sim 6$, Type-C and Type-D for $\sim 6 < N < \sim 9$ and Type-E for $\sim 9 < N$, as shown in Fig. 2. In fact, similar states to those except Type-B have already been reported in the literature[20–23]. More specifically, the recent work by Posnjak et al.[22,23] nicely demonstrated the detailed structural models for director fields in these types by combining optical observations using confocal fluorescence microscopy (CFM) and a theoretical approach based on the calculation of elastic energy. Our analysis for these types reconfirms their models, but we should note that the situation is slightly different because our droplets are not in ideal spherical shapes and are actually in spherical-cap shapes, as described later in the structural analysis.

**Topology-dependent heat-flux-driven rotation dynamics in Ch droplets.** Before describing the HIR dynamics of these topological Ch droplets dispersed in the fluorinated oligomer, we briefly examine the behaviours of the same LC material in dispersions in four typical immiscible solvents: purified water, glycerol, DMPS, and poly(perfluoro-4-vinyloxy-3- methyl-1-butene) (CYTOP, AGC Chemicals). All of these dispersions show the typical structures of Ch droplets confined by spherical boundaries with planar (water and glycerol) or homeotropic (DMPS and CYTOP) anchoring conditions[18–23]. However, none of them exhibited HIR under POM observation (Supplementary Movies 1–4). Next, we check the conventional-type HIR behaviour in the transiently formed droplets in the Iso-Ch coexistence region during phase transition in the same LC material. In this case, two types of droplets, 'Striped' and 'CC', mainly emerge during the Iso-Ch transition (Fig. 3), which is consistent with the previous study[10]. Both have a simple single-helix structure embedded in the droplet but differ from each other in the direction of the helical axis, i.e., perpendicular or parallel to the substrate plane, respectively. When the temperature gradient is applied perpendicular to the substrate, the rotational motion is induced in both types, as shown in Fig. 3 (see Supplementary Movie 5). Although their

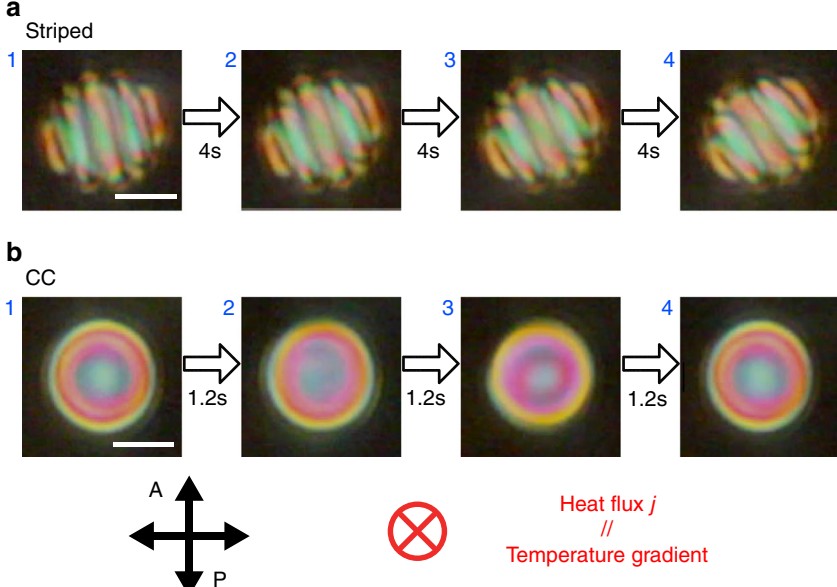

**Fig. 3** Snapshots of HIR of conventional-type droplets. The droplets are formed in the Iso-Ch coexistence region of a non-dispersed mixture of E8 and 5CB + 1.0 wt% chiral dopant. Scale bars, 10 µm. The time intervals between the snapshots for **a** and **b** are 4 and 1.2 s, respectively. The temperature is stationary at 55 °C. The heat flux is in the viewing direction, shown by the red cross symbol ($j = 6.0$ mW mm$^{-2}$)

shapes and sizes are not very stable under the temperature gradient, it can be clearly seen that the rotational speed of CC is an order of magnitude higher than that of Striped, which is consistent with the previous report[10]. Thus, HIR in the present LC material is almost identical to that in the previous reports. This result is significant in proving that even for the same Ch LC in the same cell under the same experimental conditions, the rotational speed and conversion efficiency are influenced by the arrangement of the helix inside the droplet. Therefore, this finding could further lead to an important implication that differences in the topological configuration could also influence the HIR behaviour.

We now apply a temperature gradient to the above-mentioned five topological states of the dispersed Ch droplets in the slightly miscible fluorinated oligomer. Here, rotational motion is observed for only three types, Type-C, Type-D and Type-E, as shown in Fig. 4 (see also Supplementary Movies 6–8). For Type-A and Type-B, such rotational motion is not observed by POM because their director fields are cylindrically symmetric (see the next section). Owing to the poor miscibility and high viscosity of the surrounding oligomeric medium, the dispersed Ch droplets are quite stable even under application of the temperature gradient. Amazingly, the rotational speeds of the Type-C and Type-D droplets are much higher than those of the conventional Ch droplets, as seen in the comparison between the conventional and present systems (Supplementary Movies 5 and 6). Figure 5a shows the angular velocity $\omega$ plotted as a function of the heat flux $j$. Since $\omega$ is obviously proportional to $j$, here, we define the rotational conversion efficiency as $\eta = \omega/j$. Figure 5b shows a comparison of the $\eta$ values of the dispersed Ch droplets (Type-C and Type-D) and the conventionally studied ones (Striped and CC) formed in the coexistence region during the Iso-Ch transition. We find that the $\eta$ value of the Type-C (triangles) droplet is the largest but still comparable to that of the Type-D droplets (rhomboids). On the other hand, the $\eta$ value of Striped (crosses) is several orders of magnitude smaller than the $\eta$ values of Type-C and Type-D. Note that CC (asterisks) shows intermediate values, i.e., several times smaller than Type-C and Type-D and approximately ten times larger than Striped. Thus, it is natural to conclude that HIR occurs more effectively in the Ch

droplets dispersed in the fluorinated oligomer than in the conventional droplets formed in the coexistence regions.

Figure 5c shows the $\eta$ values for Type-C, plotted as a function of temperature. Although the monotonic decrease of $\omega$ leads to small $\eta$ values in the low temperature region, the rotation itself does not stop at very low temperatures, even more than 50 °C below the Iso-Ch transition point. The temperature dependence of $\eta$ follows the Arrhenius relationship well, which implies that the system is dominated by viscosity[24]. In most of the previous studies, HIR was observed in the temperature region very close to the transition temperature, which caused it to be unclear if HIR is a transition-related phenomenon. However, the present result clearly suggests that such closeness to the transition point is neither necessary for HIR nor related to its induction mechanism. Of course, this finding also suggests that the present dispersion system is advantageous because of its ability to stabilise the Ch droplets and expand the observable temperature range down to room temperature.

The stability of the Type-E droplets is discussed based on a state diagram as a function of the droplet diameter $2R$ and the heat flux $j$ (Fig. 6). Basically, Type-E1 and Type-E2 are initially formed without a temperature gradient. By applying the temperature gradient, both start rotating while maintaining their texture, as shown in Fig. 4c, d (see also Supplementary Movie 7). An increase in the temperature gradient and, hence, a larger heat flux makes Type-E2 significantly unstable and causes chaotic motions inside the droplet (unstable state: asterisks in the diagram), while Type-E1 persists. A further increase in the temperature gradient leads to a transformation from the unstable state into Type-E3 (Supplementary Movie 8). The stabilities of these types are also discussed based on statistics of the appearances, as shown in Supplementary Figure 5. In small droplets with $2R = 15$–20 µm, Type-E1 is dominant irrespective of the applied heat flux. However, in larger droplets with $2R > 20$ µm, Type-E1 is obviously not preferred. Instead, Type-E2 becomes more dominant, especially in the low heat flux region. As mentioned above, Type-E2 is destabilised and transformed into Type-E3 through an unstable state by increasing the heat flux. This tendency is emphasised more in larger droplet sizes.

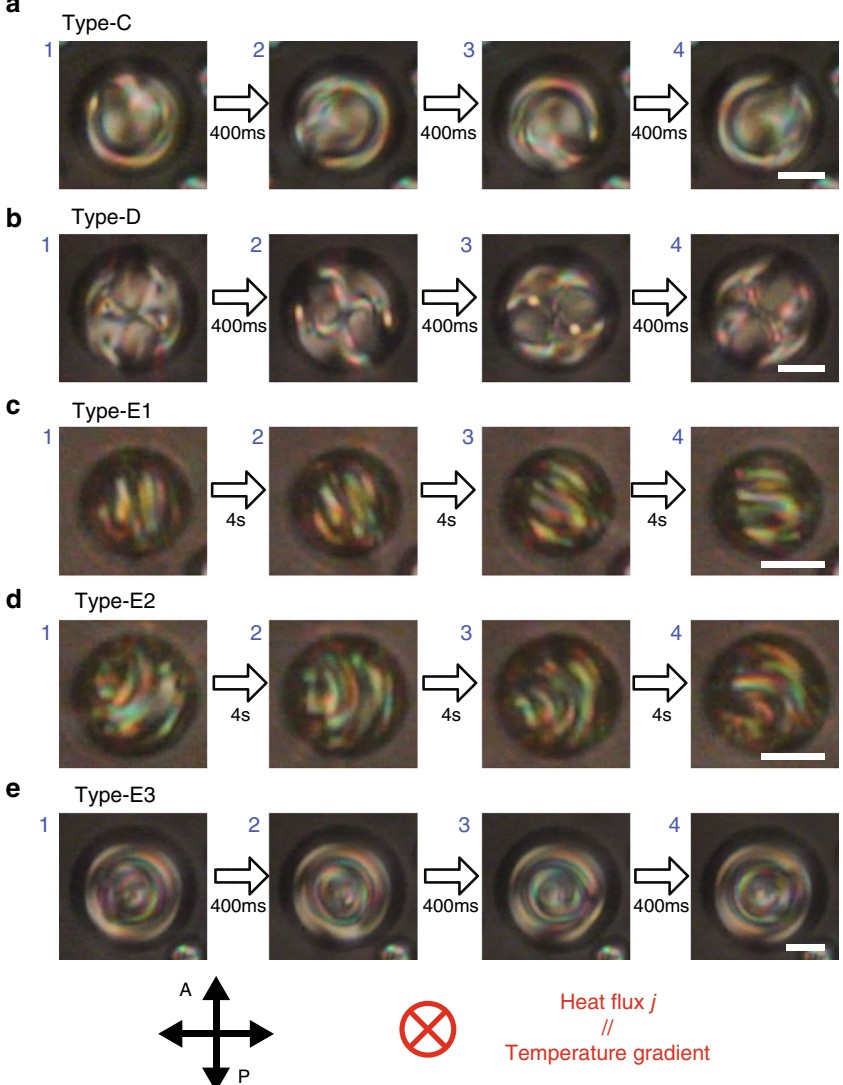

**Fig. 4** Snapshots of HIR of the Ch droplets dispersed in the fluorinated oligomer. HIR could be observed only in these five types. Scale bars, 10 μm. The time intervals between the snapshots for **a** Type-C, **b** Type-D and **e** Type-E3 are 400 ms and for **c** Type-E1 and **d** Type-E2 are 4 s. For all types, the temperature was stationary at 55 °C. The heat flux was applied in the viewing direction, as shown by the red cross symbol ($j = 6.0$ mW mm$^{-2}$ for Type-C, Type-D and Type-E3, and $j = 1.5$ mW mm$^{-2}$ for Type-E1 and Type-E2). The concentration of the chiral dopant was 1.0 wt% for Type-C, Type-D and Type-E3 and 2.0 wt% for Type-E1 and Type-E2

Figure 7 shows the scatter plots of $\eta$ in Type-E with various strengths of chirality as a function of $2R$. In all three types, clear differences can be seen among the different strengths of chirality. For Type-E1 and Type-E2, $\eta$ decreases as $2R$ increases if the strength of the chirality is fixed. If $2R$ is fixed, then a lower $\eta$ is obtained for stronger chirality. On the other hand, the behaviour in Type-E3 is the opposite; i.e., $\eta$ increases with either $2R$ or the strength of chirality. Interestingly, $\eta$ in Type-E3 is larger than that in the other two cases, and thus, HIR in Type-E3 is more efficient.

From these results, it is obvious that $\eta$ in the present dispersion system is much higher than that in the previous reports for droplets formed in the Iso-Ch coexistence region. One speculation deduced from this fact is that HIR in the current system is driven by a heat-flux-induced mass-flow of a small amount of oligomer molecules penetrating into the Ch droplets, analogous to the flow-induced rotation in smectic-C* reported by Tabe et al.[25]. This hypothesis is intuitively consistent with the above

result that shows no rotation in the dispersions using the immiscible solvents. However, to make this conclusion, we still need further verification of the intermediation of such a material flow, for example, by using the nanoparticle tracking technique.

At the same time, it would also be worth discussing whether the observed HIR is rotational mass motion (rotational flow or rigid-body rotation) or simple director rotation without any mass transport. If we focus our attention on one of two or more droplets that gather in the close vicinity of one another, their rotation speed is almost the same as that of standalone droplets of the same type of the same size. Additionally, their centres of mass are fixed, as seen in Supplementary Movies 6–8. These findings mean that there are likely no frictional forces between them. This circumstance is possible if the rotational mass motion is negligibly small or completely non-existent[10,26]. Of course, this suggestion is only a speculative deduction from the observational facts and still requires further analysis by extensive studies using both theoretical and experimental approaches.

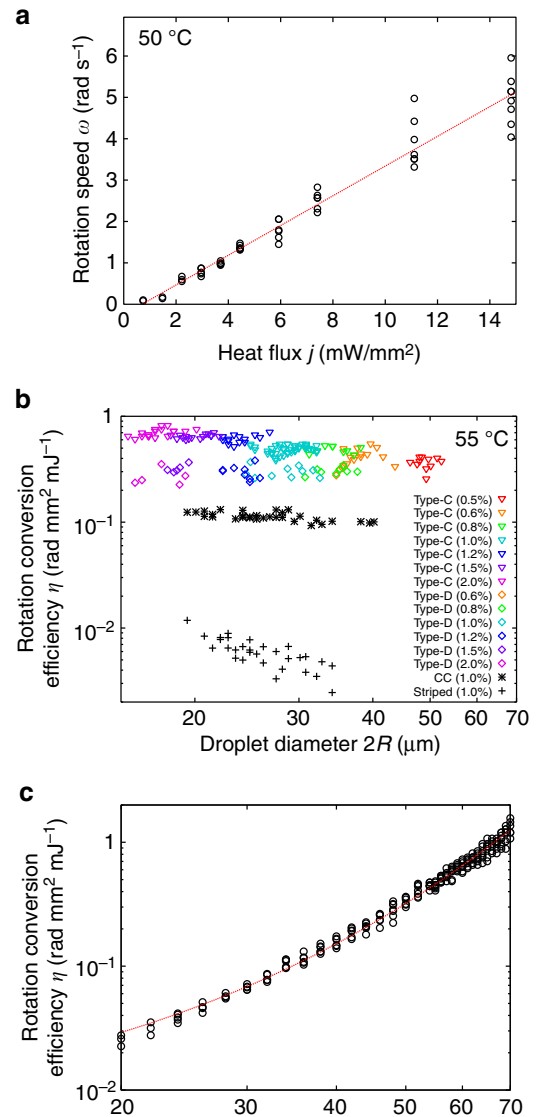

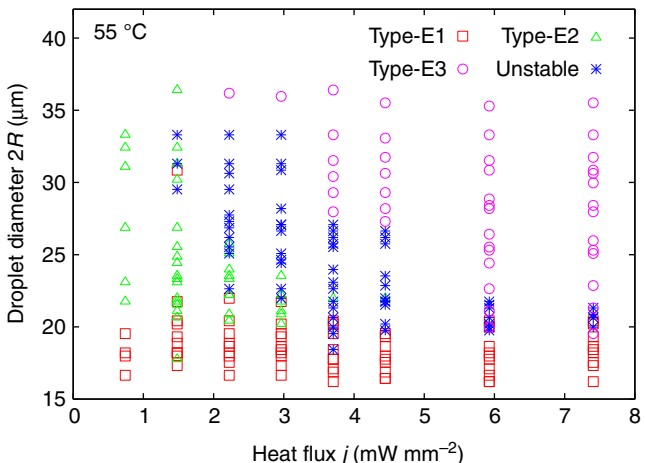

**Fig. 6** State diagram of Type-E under the temperature gradient. The droplet sub-types are plotted in the plane of ($j$-$2R$). The chiral content was 2.0 wt %. In the small $j$ regions, only Type-E1 (squares) and Type-E2 (triangles) appear and show HIR without texture change (Video 4a in Supplementary Materials). However, by increasing $j$, Type-E2 is destabilised and transformed into Type-E3 (circles) (Supplementary Movie 8). In the region between Type-E1/E2 and Type-E3, a chaotic turbulent texture was continually observed (unstable state, asterisks)

CFM images for the Type-A droplet, which clearly show the radial director distribution of Type-A (Figs. 1o and 8a)[17]. This arrangement is reasonable in the present case because the interface between the LC and the oligomeric solvent has a homeotropic anchoring property[20]. Note that the side-view cross-sectional CFM image suggests a spherical-cap shape, which means that the droplet slightly touches the top or bottom substrate with a finite contact angle. Actually, in the present study, this situation is common in all five types, Types-A–E.

On the other hand, Type-B appears for $\sim 2 < N < \sim 6$. The POM image of Type-B also shows an extinction cross, but with slight swirling in a single direction, depending on the chirality (Fig. 1b). The cross-sectional CFM observation (Supplementary Fig. 1b) suggests that the nematic director in Type-B is twisted along the viewing direction (=cell normal) and hence is not spherically but cylindrically symmetric. This reduction of symmetry is attributed to the spherical-cap shape of the droplet. Based on these considerations, the nematic director field of Type-B can be modelled as shown in the cross-sectional schematics in Figs. 1p and 8b. We should note that both in Type-A and Type-B, due to this structural rotational symmetry about the rotation axis, the director rotation cannot be observed by POM even if HIR occurs.

To ensure these director models, we simulated the POM textures by using the Jones matrix calculation on numerically formulated director structures (Supplementary Note 1). The calculated POM images reconstruct the POM observations well (Fig. 1h, i) and hence confirm the validity of our structural modelling and numerical formulation for Type-A and Type-B.

**Type-C with a helical structure having a surface point defect.** Type-C is observed when $\sim 6 < N < \sim 9$, with the typical appearance of a 'U-like' birefringence pattern. In fact, similar droplets and their internal director field have been reported in literature[22,23]. Supplementary Figure 2 shows the cross-sectional CFM images of the Type-C droplet upon HIR. The images taken near the substrate surface ((iv) in Supplementary Fig. 2) resemble only those of Type-A, and thus, HIR is not confirmed because of the radial symmetry. On the basis of these cross-sectional CFM

**Fig. 5** Behaviour of HIR. **a** Linear dependence of the rotational speed $\omega$ on the applied heat flux $j$ in Type-C (25–32 µm diameter) with 1.0 wt% chiral dopant. **b** Rotation conversion efficiency, $\eta = \omega/j$, plotted as a function of the droplet diameter $2R$ for Type-C, Type-D, striped type and CC type. See legends for the chiral content. **c** Temperature dependence of $\eta$ in the Type-C droplets with $2R = 25$–$32$ µm. The chiral content was 1.0 wt%. Note that the Iso-Ch transition temperature was 72 °C. As shown by the red dotted curve, the experimental data will follow the Arrhenius equation, $\omega/j = ae^{-\frac{b}{T+273}}$[24], where $a = 4.78\times10^9$ rad mm² mJ⁻¹ and $b = 7570$ K

**Topologies and optical simulations of dispersed Ch droplets.** Before discussing the possible origin of the differences in these HIR characteristics among the topological types, we analysed their internal nematic director field to specify the topological and geometrical configurations. This work was performed complementarily by using POM, CFM and optical calculation with the Jones matrix method.

**Type-A and Type-B having one central point defect.** As shown in Fig. 2, Type-A emerges only for $N < \sim 2$, with a typical appearance of an extinction cross under POM, which suggests the existence of radial symmetry around the central axis. Supplementary Figure 1a is an example of the typical cross-sectional

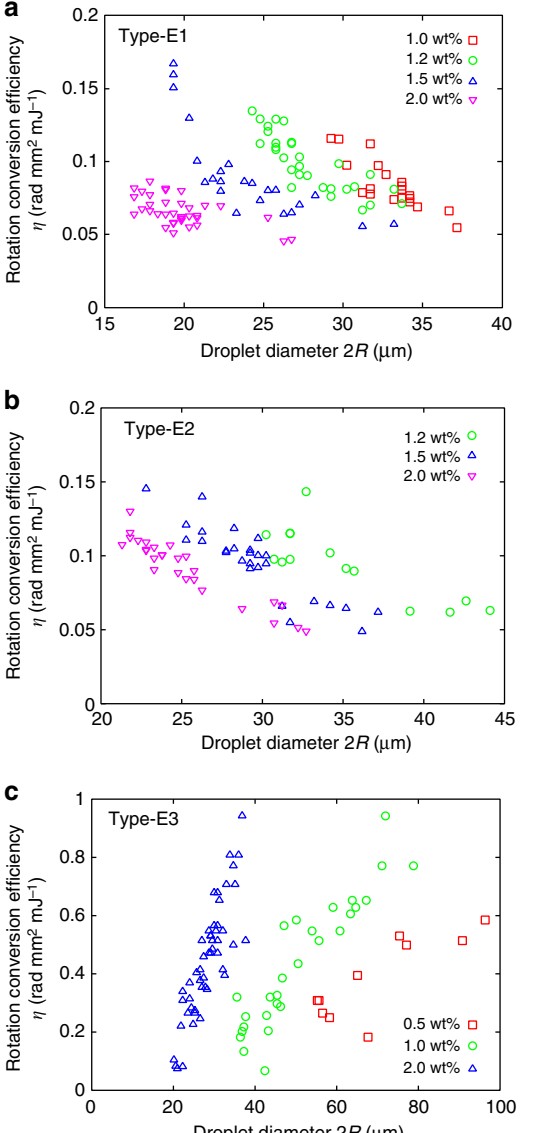

**Fig. 7** Rotation conversion efficiency $\eta$ of Type-E. A clear difference is observed among the samples with different strengths of chirality (see legend). **a**, **b** Type-E1 and Type-E2 show $\eta$ decreasing as either $2R$ or the chiral content increases. On the other hand, the behaviour in **c** Type-E3 is completely opposite

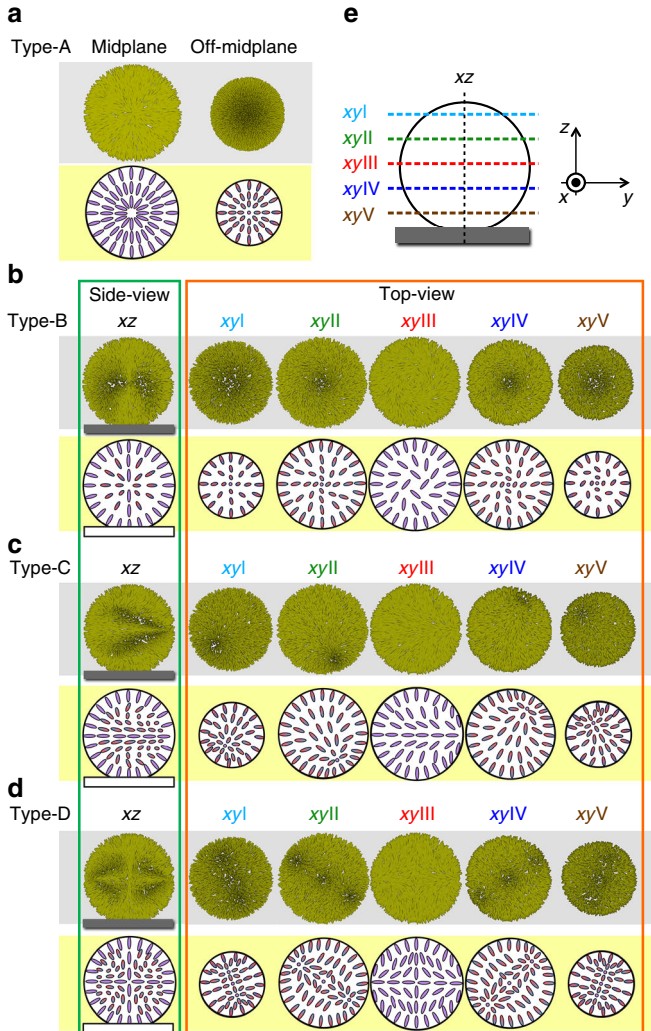

**Fig. 8** Chart of the schematic models of the director fields for Types-A–D. **a** Type-A, **b** Type-B, **c** Type-C, and **d** Type-D. The upper rows are the numerically depicted nematic directors at certain cross-sectional positions using the trial functions, with Supplementary Equations (1) and (2) for Type-A and Type-B, Supplementary Equations (3) and (4) for Type-C, and Supplementary Equations (5) and (6) for Type-D. The lower rows are the simplified cartoons of the corresponding director field using ellipsoids to show the point-by-point distributions of the directors projected onto the plane. The colour gradation of the ellipsoids from red to blue represents the director tilt from up to down against the plane. For Type-A, only two horizontal cross-sections at the midplane and off-midplane are shown. For each of the other three types, one vertical ($xz$) and five different horizontal ($xy$I–$xy$V) cross-sectional positions are chosen according to the schematic (**e**). Note that $xy$III corresponds to the equatorial plane (=midplane)

images, a possible model of the director field of Type-C is proposed, as shown in the schematics in Figs. 1q and 8c. Each $xy$ cross-section in Fig. 8c has only one singular point away from the centre. These singular points have a vertical director orientation except at the equatorial plane, at which a point defect appears on the surface (hence on the equator) ($xy$III in Fig. 8c). By tracing these singular points, it can be found that they are arranged in a spiral manner; i.e., a one-dimensional helicoid around the central axis is formed in the Type-C droplet. Although the materials and experimental conditions used in the present study are different from those in the studies by Posnjak et al.[22,23], the results imply almost the same director structure except for the spherical-cap shape of our droplets.

Numerical simulation of the POM texture for Type-C is conducted based on the formulation of the above director structure (Supplementary Note 2). As can be clearly seen in Fig. 1c, j, the POM images simulated using this director field

reconstruct the POM image well. Of course, this simulation strongly supports the validity of the director model for Type-C. Note that Type-C always appears with a texture similar to that in Fig. 1c, which means a fixed orientation with winding in the viewing direction, as shown in Fig. 8c.

**Type-D with a double helix having three point defects**. Type-D appears in almost the same range of $N$ as Type-C, and a similar state has also been reported in the literature[22,23]. The cross-sectional CFM images of the Type-D droplet upon HIR are shown in Supplementary Fig. 3. Figures 1r and 8d are the structural model for the director field based on this observation.

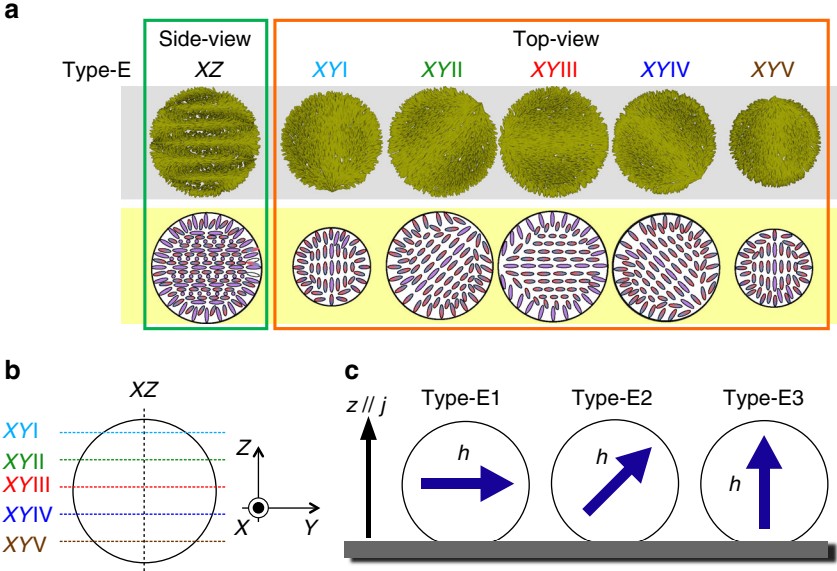

**Fig. 9** Models of director fields for Type-E. **a** Schematics of the director models of Type-E. See the caption of Fig. 8 for explanations of the drawings. The trial functions, Supplementary Equations (7) and (8), are used for the numerical depiction of the director, with the sample coordinates ($X,Y,Z$) introduced by setting the $z$-axis along the helical axis $h$. Six cross-sectional positions ($XZ$, and $XY$I–$XY$V) are chosen according to the schematic (**b**). Note that $XY$III corresponds to the equatorial plane (=midplane). **c** Geometrical configuration of the helical structure embedded in the droplets attached on the substrate. According to the simulated polarised micrographs, by using this director field for Type-E (Fig. 1l–n), it was concluded that the difference among the three types of Type-E was attributed to the directional difference of the helical axis $h$ with respect to the substrate normal. Note that the $z$-axis in **c** corresponds to the cell normal, which is parallel to the temperature gradient and, hence, the heat flux $j$

Each $xy$ cross-section in Fig. 8d has three singular points—one at the centre and the other two arranged symmetrically on either side. By tracing these singular points, it is recognised that the outer two are arranged in a spiral manner similar to Type-C, i.e., a double helix structure winds around the central axis in the Type-D droplet. At the equatorial plane, the outer two are exposed to the surface and hence are located on the equator as surface point defects. Using optical simulations under the formulation of this director field (Supplementary Note 3), we successfully reconstruct the POM images for Type-D (Fig. 1d, k). It should be emphasised that our analysis for Type-D is again reasonably in accordance with the previous report by Posnjak et al[22,23]. Similarly to Type-C, the appearance of Type-D indicates a fixed orientation, as shown in Fig. 8d, such that Type-D also twists along the viewing direction.

**Type-E with a surface coiled defect structure.** Type-E with complex textures with streaks appears at ~ 9 < N. As mentioned in the previous section, this type can be further divided into three subgroups, Type-E1 (long-pitch streak pattern), Type-E2 (short-pitch streak pattern), and Type-E3 (concentric streak pattern). The cross-sectional CFM images of the Type-E1 droplet show dark fringes that rotate with HIR (Supplementary Fig. 4). Unlike the other types, the images near the substrate also show fringes with rotation (iv in Supplementary Fig. 4). This finding suggests that the director periodically twists along the direction of the fringes. Thus, it is natural to consider that a single-helix structure of the Ch LC is embedded in and laid onto the substrate plane. However, it is still curious that in fact, the rotating droplet never shows extinctions, i.e., the droplet shows clear fringes even at the rotation angles that correspond to the fringes being perpendicular (Supplementary Fig. 4b) to the incident polarisation. If the director structure is a simple helix, the fluorescence for this angle should be very weak. The existence of a second twist in the

viewing direction is a plausible reason for this inconsistency. In fact, such a three-dimensionally twisted director configuration has already been observed in some Ch LC systems confined between homeotropic surfaces[27–29]. Based on these considerations, we model a three-dimensionally twisted structure for Type-E1 (Fig. 9a). Each cross-section has only a single singularity at the surface. This arrangement is a representation of a vortex-like topological defect that winds around the droplet surface, which has already been predicted theoretically[20]. Although the presence of such a vortex defect is almost ensured by the director distribution, the present POM and CFM images are too complicated and insufficiently clear to recognise such a defect structure.

Based on the formulation of the above structural model (Supplementary Note 4), POM images are simulated for three different configurations of the direction of the main helical axis $h$ with respect to the laboratory coordinates($x,y,z$), i.e., $h \perp z$, $h \parallel z$, and the inclined state, as schematically shown in Figs. 1s–u and 9c. The calculated image for $h \perp z$ simulates the POM of Type-E1 well (Fig. 1e, l), which confirms the validity of the structural model and configuration. Interestingly, the simulated results for $h \parallel z$ and the inclined state (Fig. 1g, f, respectively) show a close resemblance to the POM images of Type-E3 and Type-E2 (Fig. 1n, m, respectively). Thus, we conclude that the assumed director field for Type-E is basically reliable, and the textual difference among the three types of Type-E is due to the difference in the direction of the main axis $h$ with respect to the $z$-axis.

**On the relation between the structure and the rotational conversion efficiency.** On the basis of our experimental observation and analysis, the structural properties and the physical features of HIR in the observed 5 types of Ch droplets are summarised in Fig. 1. Type-A and Type-B have the radial director configurations observed in typical N and Ch droplets[17,22,23] and, hence, are

identical from the viewpoint of topology. As for Type-C and Type-D, although both types have a twist of the ranging singular points along the cell normal as mentioned above, they are not topologically homeomorphic. The former has one surface point defect with a +1 topological charge, while the latter has one central point defect with a −1 topological charge and two surface point defects with a +1 topological charge. Type-E is also a unique topological state that has a surface coiled defect surrounding the droplet, which is assumed to be a state similar to that predicted by the numerical calculation in ref. [20].

On the other hand, in the previous work for Ch droplets in the Iso-Ch coexistence region, HIR was compared in terms of the direction of the helical axis with respect to that of the applied temperature gradient, and it was found that the rotation is generated more efficient when the two directions are in parallel[10,11]. Thus, even in the present work, the rotation behaviour is discussed in relation to the geometrical configuration of the helical structure. For Type-C, for our design of its director field, the droplet has a one-dimensional helical structure with the helical axis parallel to the temperature gradient (Fig. 8c). Similarly, Type-D is also considered to have a one-dimensional helical structure in the direction of the temperature gradient (Fig. 8d). Therefore, efficient HIR is reasonably induced in these two types.

The situation for Type-E is somewhat more complicated. As shown in the stability diagram (Fig. 6) and the statistics chart (Supplementary Fig. 5), Type-E2 is transformed to Type-E3, depending on the heat flux. According to the structural analysis, this transformation is attributed to the directional change of the helical axis with respect to the direction of the applied temperature gradient. In the case where the applied temperature gradient is small enough, the helical axis prefers to be parallel to or inclined from the z-axis to minimise the sum of the anchoring and elastic energies. The above structural analysis suggests that these configurations correspond to Type-E1 and Type-E2, respectively. On the other hand, if the temperature gradient increases, the helical structure is changed to make its helical axis parallel to the direction of the temperature gradient (transformation to Type-E3). This arrangement would occur simply because the helical axis prefers to orient along the direction of diffusion, similar to the free-yaw behaviour of wind turbines, as will be discussed elsewhere. As far as we observe in the present study, this transformation into Type-E3 occurs especially for Type-E2 and not for Type-E1, perhaps because Type-E2 is energetically less stable than Type-E1. As mentioned above, the rotational conversion efficiency $\eta$ values in Type-E1 and Type-E2 show the same trend (Fig. 7a, b); i.e., $\eta$ decreases as the strength of chirality increases, which is in fact similar to the striped type formed in the Iso-Ch coexistence region of the present material and a previously reported system[10]. At the same time, for Type-E3, the trend in the dependence of $\eta$ on the strength of chirality is completely different: $\eta$ increases as the strength of chirality increases. In fact, this trend in Type-E3 is also similar to that in CC in the Iso-Ch coexistence region in the present material and a previously reported system[10]. Obviously, these behaviours are associated with the directional configurations of the helical axis, regardless of whether it is parallel or not to the direction of the temperature gradient. In addition, it is important to remember that $\eta$ in Type-E3 is far higher compared to that in Type-E1 and E2. The same tendency is also found in the relationship between Stripe and CC (Fig. 5b). However, in spite of the same trend, $\eta$ in Type-E3 in the present dispersion system is very high, even among the previously reported systems.

In summary, the behaviour of HIR, especially $\eta$, depends on not only the LC material and direction of the helical axis but also the topology of the system. For example, Type-C, Type-D, and

Type-E3 have the helical axis parallel to the direction of the applied temperature gradient. However, their absolute values of $\eta$, chirality and droplet-size dependence are quite different from one another, as shown in the experimental data (Figs. 5b and 7c). These differences are obviously related to the structural and topological features, i.e., one central point defect, one central and two surface point defects, and a surface coiled defect in the Type-C, Type-D and Type-E3 droplets, respectively. Nevertheless, it would be worth noting that it is still unclear whether or how the surface wetting at the contact with the substrate affects HIR in such spherical-cap geometries. For example, the modified flow fields at the contact line might influence the effect. Although we still need to perform extensive experimental and analytical studies, such a connection between the HIR and topology is very interesting and must be key to understanding the mechanism further.

## Discussion
We find that HIR could be induced in Ch droplets dispersed in an oligomeric solvent. Confinement in a spherical-cap geometry surrounded by homeotropic surfaces stabilises the Ch droplets in five topological states, which are referred to as Types-A−E, depending on the droplet size and the strength of chirality, as confirmed by POM (Fig. 1). Steady director rotation is induced by a temperature gradient only in Type-C, Type-D and Type-E. The rotational conversion efficiencies of Type-C and Type-D are several orders of magnitude higher than those of the previously known HIR in the Iso-Ch coexistence region, and interestingly, Type-E showed further acceleration. Because of the binary separation, HIR in these dispersed Ch droplets can be realised far below the I-Ch transition point, most notably in the room temperature range. Thus, such a dispersion form is considered to be a more suitable and efficient system for HIR.

With the aid of CFM, we model plausible director structures for these five topological types and confirm their validity by simulating POM images using the Jones matrix method. The origin of the differences in the rotational behaviour among the droplet types is discussed in terms of these structural models and the relative direction of the applied temperature gradient. It is also revealed that Type-E under a temperature gradient shows a directional transformation through the competition between the anchoring and elastic free energies, in addition to the torque applied to the Ch structure, similar to the free-yaw behaviour of solid turbines.

The present results suggest that further optimisation is possible for the HIR phenomenon by selecting adequate material combinations, such as those for stabilisation, speed-up and efficiency, depending on the purpose of the experiment. However, further experimental and analytical studies are necessary to understand the physical origin and mechanism of the HIR phenomenon.

## Methods
**Sample preparation**. The cholesteric (Ch) liquid crystal (LC) was prepared by adding a chiral dopant, (S)-2-Octyl 4-[4-(Hexyloxy)benzoyloxy]benzoate (S811, TCI), to a host nematic (N) liquid crystalline mixture of E8 (Merck) at a concentration of 0.2–2.0 wt%. We used a fluorinated oligomer (PF656, Omnova Solutions Inc.) as an isotropic (Iso) solvent. The viscous modulus of PF656 is 1.8 Pa s at room temperature and is 1−2 orders of magnitude larger than the modulus of a LC, 4-cyano-4'-pentylbiphenyl (5CB)[30]. The phase diagram of the mixture of PF656 and E8 shows wide Iso-N and Iso-Iso coexistence (phase-segregated) regions, while they are well-mixed when the E8 content is less than ~ 10 wt% or greater than ~ 90 wt%, as shown in Supplementary Fig. 6. Thus, to make the droplets well-dispersed in the Iso solvent, we set the content of the Ch LC in PF656 to be ~ 20 wt%. The dispersion of the Ch droplets was injected into 50-μm-gap cells that consisted of two clean glass substrates (~ 1.35 mm thick) without any surface treatment.

As reference samples in the dispersion systems with immiscible solvents, we used purified water, glycerol, DMPS (Wacker Chemie AG), and poly(perfluoro-4-

vinyloxy-3- methyl-1-butene) (CYTOP, AGC Chemicals) as the solvents. Mixing the solvents and the above-mentioned Ch LC with the weight ratio of 19:1, we made the dispersion systems.

As a reference sample of HIR in the Iso-Ch coexistence state, we used a mixture of E8 and 5CB (TCI) at a weight ratio of 3:2 for a host N LC. The concentration of the dopant was 1.0 wt.%, and the phase sequence of this reference Ch LC was Ch;54°C;Iso+Ch;56 °C;Iso. More details are available in ref. [10].

**Temperature and gradient control**. The temperature and temperature gradient were precisely controlled by a combination of a homemade furnace and a commercial thermo plate (Tokai Hit Co.), for which the heater outputs were regulated by a proportional-integral-differential controller with thermocouple temperature sensors. The sample cell was sandwiched between these two stages having different temperatures, as schematically shown in Supplementary Fig. 7.

**Polarising microscopy and optical measurements**. For the POM, we used a commercial polarising microscope (BH2, Olympus) equipped with a digital camera (EOS Kiss X50, Canon). POM was used not only for texture and HIR observations but also for determination of the helical pitch and birefringence, which was required for the optical calculation. To evaluate the helical pitch, the Grandjean-Cano method was utilised for wedge-shaped homogeneous cells, which were fabricated using two glass plates with a 100-μm-thick PET (polyethylene ter-ephthalate) spacer film. Birefringence was deduced from transmission spectra measured for 20-μm-thick homogeneous cells with a commercial spectroscope (USB2000+VIS-NIR, Ocean Optics)[31,32]. For both the pitch length and birefringence measurements, polyimide (AL1254, JSR) was used as a homogeneous alignment material that coated the cell substrates. See Supplementary Fig. 8 for the experimentally obtained dispersion curve of birefringence.

**Optical simulation with the Jones matrix method**. The Jones matrix method was used for simulating POM images[33]. The refractive indices used for PF656 for ordinary light, E8 for ordinary light and E8 for extraordinary light at the sodium D-line wavelength are 1.39, and 1.53 and 1.76, respectively[34]. The helical pitch and the birefringence used for the calculation were obtained by optical measurements for bulk in LC cells using POM (see above). The droplets were assumed to be a stack of 100-nm-thick slabs. The calculation was conducted for flat-intensity light with 22 different wavelengths, and the output was used to obtain RGB images using colour-matching functions.

**Confocal fluorescence microscopy**. For the CFM, the fluorescent dye n,n'-bis(2,5-di-tert-butylphenyl)-3,4,9,10-perylenedicarboximide (BTBP, Sigma-Aldrich) was doped into the Ch mixture with a concentration of ~ 0.02 wt%. BTBP is known for its linear dichroism property. Since BTBP tends to be aligned along the local LC director field, its fluorescence is pronounced in the regions where the nematic director is parallel to the polarisation of the incident excitation light[35]. In addition, localisation of the dye molecules at the defects works for identification of the defect positions[36,37]. CFM was performed with an inverted microscope equipped with an excitation laser (TCS SP8, Leica). The excitation wavelength was 488 nm, which is where BTBP absorbs. We used a water-immersion objective lens with a numerical aperture of 1.2, which has a focal depth of ~ 1.0 μm.

**Data availability statement**. All data that support the findings in this study are available in the article and in Supplementary Materials. Additional information is available from the corresponding author upon reasonable request.

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

## Acknowledgements

Authors thank Ms. Rina Mogaki and Prof. Takuzo Aida for the use of their lab rooms and facilities for the confocal microscopy. This work was partially supported by JSPS KAKENHI Grant Number 15K17739.

## Author contributions

J.Y. and F.A. designed the study. J.Y. performed experiments. J.Y. and F.A. analysed data and prepared the manuscript. All authors discussed the results and approved the final manuscript.

## Additional information

**Competing interests:** The authors declare no competing financial interests.

