## [Peer Review File · Nature Communications]

Reviewers' comments:

Reviewer #1 (Remarks to the Author):

This work reports on Lehman rotation of chiral nematic microdroplets of different topology, induced by the temperature gradient and transport of heat across the droplets dispersed in an isotropic medium. An extensive analysis was performed including the influence of the chemical composition of the carrier medium and different internal structures of droplets. It was found that Lehman rotation occurs only in very specific and quite viscous carrier fluid, based on fluorinated material. This is quite surprising and the reason for that remains unclear. The surface boundary condition for the combination of LC-carrier is homeotropic. The authors found that Lehman rotation of droplets is observable in a very broad temperature range. Most surprisingly, they found that the efficiency of heat-flux induced rotation depends on the internal structure of droplets, their topology and is characterized by the ratio of the pitch to the radius of the droplet.

The subject of this paper is interesting and timely, because it tackles the aspects of topology of chiral nematic objects and addresses the connection between their physical properties to topology. The experiments are well designed and performed, the results seem to be free of flaw. The conclusions are sound.

While the subject of this work is novel and certainly interesting for the readership of this journal, the manuscript needs significant revision before any conclusion on acceptance could be made. What I find redundant in the present manuscript is the extensive FCPM analysis of the internal structure of droplets, which is not new. Namely, Posnjak et al. (cited as reference 16) have recently presented in Nature Communication paper a very precise and decisive FCPM analysis of the internal structure of homeotropic droplets that are identical to the droplets presented in this paper. I suggest the paper be restructured and rewritten so the FCPM analysis, which does not give new results, be moved to Supplementary material. What I am actually missing in the present manuscript, are the answers to the following questions:

1. Is the rotation driven by the transport of energy (heat) or mass? The sensitivity of Lehman rotation of chiral nematic droplets to the chemical structure of carrier fluid indicates it is transport of mass rather than energy. Fluorinated solvent is partially miscible with LC, meaning the oligomeric molecules could flow through the LC droplets. I remember the work of Yuka Tabe on the mass-flow induced rotation of ferroelectric smectic-C* layers that should be mentioned here and analysed.
2. Does the efficiency of rotation depend on the orientation of the symmetry axis of the droplet (if any) with respect to the direction of heat flow? Does the droplet reorient and stabilizes in certain orientation?
3. Is there any relation between the number of topological point defects and the efficiency of Lehman rotation?

In addition, I have some technical comments:

- 1- Page 2, line 3: HIR of what?
- 2- Page 2, line 23: "more efficient than in the bulk" needs explanation and better description.
- 3- Page 3, lines 15-18. The authors should state more clearly that they performed experiments in other solvents and no rotation was observed.
- 4- Page 4 line 9: early work on LC droplets is not cited, such as Volovik and Lavrentovich, and Crooker et al.
- 5- Page 4 lines 10 -20: these structures were reconstructed recently by Posnjak et al. Using new method of director reconstruction from FCPM data, references 15 and 16. This should be stated in the revised manuscript.
- 6- Page 6, lines 8-10: what is the orientation of the symmetry axis of droplets with respect to heat

flow direction?

7- Page 7, line 13: "chaotic motion inside". Inside what, droplet?

8- Page 14, line 8-9: how can you see the rotation of type-C structure if the direction of the helical axis is along the heat flow direction?

9- Optical indices should be given for all materials mentioned in Materials and methods section.

Reviewer #2 (Remarks to the Author):

This is a very nice piece of work on heat-induced rotation of director structures in chiral liquid crystal droplets, aka Lehmann rotation, that combines careful and thorough experimental study together with modelling efforts. The peculiarity of this work is to address the relationship between the three-dimensional chiral director field and the thermally induced rotational effect, that is found to be robust in the used arrangement. Technically, this has been possible owing to the use of a specific host fluid allowing the bulk director field to "slide" inside the spherical interface between the cholesteric and the surrounding fluid. The fact that droplets nearby rotating droplets have fixed center of mass even (as one can see from videos) emphasize the absence of rigid-body rotation, a point that would deserve some comments. Also, I recognize the efforts made in constructing ansatz for the director field and I think that they will feed constructively further discussions on the topological features of confined chiral nematic systems. Still, in view of the proposed director field for the type-E droplets, it might be a useful information to discuss the statistics of appearance of sub-types E1-2-3 vs expected structural stability. Also, regarding the type-B that was not reported in previous study dealing with droplets with homeotropic boundary conditions, some discussion is missing. Indeed, this seems to happen because the system is actually different, hence there is less fundamental surprise than it appears in the text. In fact, here the droplets sit on substrate, which modifies the 3D confinement, hence the release of a twisted structure as the ratio p/R decreases. In short, strictly speaking, one cannot compare topological diversity of previous works with the present one, and this should appear clearly. This does not weaken the interest of the proposed spherical ansatz for the droplets.

Summarizing, I encourage the authors to bring additional comments on the few raised points above and, basically, I am supporting the publication of this well-written manuscript. It brings valuable information on a non-trivial phenomenon that is found to be refreshed by recently unveiled topological diversity in droplets of cholesterics having perpendicular surface boundary conditions for the director.

Point-by-Point Replies to Reviewers

To Reviewer 1

Thank you very much for your invaluable opinions and suggestions. We are also grateful to your constructive feedback to our manuscript. Based on the comments by the reviewers and the editor, we made a major revision on the manuscript. There are so many revisions and corrections in the text, as well as modifications of figures and Supplementary Materials. Besides, we address the answers to the questions raised by the reviewers one by one as follows. We believe the current version of the manuscript may be satisfactory to you. Your positive decision would be greatly appreciated.

1. What I find redundant in the present manuscript is the extensive FCPM analysis of the internal structure of droplets, which is not new. Namely, Posnjak et al. (cited as reference 16) have recently presented in Nature Communication paper a very precise and decisive FCPM analysis of the internal structure of homeotropic droplets that are identical to the droplets presented in this paper. I suggest the paper be restructured and rewritten so the FCPM analysis, which does not give new results, be moved to Supplementary material.

=> Thank you very much for your suggestion. Needless to say, we recognize the great effort and preciseness of the FCPM (CFM) analysis in that paper by Posnjak et al. (Ref.16 of the previous version of the manuscript). Although the materials and experimental conditions used in our study are different, some of our results are reconfirmation of their results by showing almost the same director models. So we should admit that these are not necessary to be shown in our present manuscript. However, we still think that these director fields should appear in the main text, because it is required to discuss the relationship between the director structure and the heat-flux-induced rotation (HIR). Moreover, the conditions for appearances of droplet types are different from the previous works. Thus, we moved the detailed description including the CFM images out to new Section 3 in Supplementary Materials, but still we left some schematic images of the analyzed director fields (new Figures 8 and 9) in the main text. Accordingly, we made reorganization on the manuscript.

2. Is the rotation driven by the transport of energy (heat) or mass? The sensitivity of Lehman rotation of chiral nematic droplets to the chemical structure of carrier fluid indicates it is transport of mass rather than energy. Fluorinated solvent is partially miscible with LC, meaning the oligomeric molecules could flow through the LC droplets. I remember the work of Yuka Tabe on the mass-flow induced rotation of ferroelectric smectic-C* layers that should be mentioned here and analysed.

=> This essential question has been discussed for a long time in the series studies on HIR in cholesteric LC droplets. We agree with you that most presumably, the driving force is some kind of mass flow induced by the temperature gradient. The fact that the rotational conversion efficiency depends on the miscibility of the surrounding medium to the LC materials may support this hypothesis, because the efficiency seemingly depends on the flowing molecules dissolved in the droplet as you point out. However, to justify this hypothesis, it still needs further proving the existence of such a flow, as well as the clarification of the relationship between the flow and the HIR phenomenon itself. This

would be possible by extensive studies with quantitative evaluation of the flow and theoretical modeling of that. Of course, these are challenging tasks but worth tackling, and in fact some experiments are on-going now, but at this moment we have only few things to put on the table in this manuscript. To discuss this point, we added some sentences with a reference to the work by Tabe et al. as Ref.25, as follows.

Page 8, Line 22

“From these results, it is obvious that η in the present dispersion system is much higher than in the previous reports for the droplets formed in the Iso-Ch coexisting region. One speculation deduced from this fact is that HIR in the current system is driven by a heat flux induced mass-flow of a little amount of oligomer molecules penetrated in the Ch droplets, analogous to the flow-induced rotation in Smectic-C* reported by Tabe et al.[25] This hypothesis is intuitively consistent with the above result showing no rotation in the dispersions using the immiscible solvents. However, to conclude so, still we need further verification of the intermediation of such a material flow, for example, by the nano-particle tracking technique.”

(New Reference)

[25] Y. Tabe and H. Yokoyama, *Nat. Mater.*, 2003, 2, 806.

3. Does the efficiency of rotation depend on the orientation of the symmetry axis of the droplet (if any) with respect to the direction of heat flow? Does the droplet reorient and stabilizes in certain orientation?

=> To begin with, it would be worth noting that the present droplets are not really spherical but spherical-cap-like, because they are touching the cell substrate with a finite contact angle, as mentioned in the text. In the case of the Type-B~D droplets, the rotational-symmetry axis (Type-B and D) or winding axis (Type-C) is fixed along the substrate normal (=z-axis // heat flow) as shown in new Fig. 8 (old Fig. 6). In these cases, they don't change their orientations even under the heat flow. On the other hand, Type-E droplets reconfigure their orientations with respect to the direction of the heat flow. More specifically, only the initially-generated Type-E2, whose helical axis is tilted from the substrate normal, shows transformation into Type-E3 by aligning the helical axis along the substrate normal. Interestingly, Type-E1 doesn't change, maybe because of less energetic penalty in anchoring and elastic deformation. Thus, the answer to your question is 'yes-and-no', i.e. the stabilization and orientation of droplets depend on the type of the droplets. Again, reorientation of the helical axis by the heat flow was observed only in Type-E2 transforming into Type-E3. To describe this more clearly, we revised the manuscript. In particular, Fig. 1 was completely reconstructed to present the relationship between the structural and geometric features and HIR of droplets.

4. Is there any relation between the number of topological point defects and the efficiency of Lehman rotation?

=> In the present work, it can be seen in the difference of the rotation efficiency among Type-C, D and E3 as shown in new Figs. 5b and 7 (old Figs. 3b and 4). Since all of

these three types have the helical axis parallel to the heat flow, we consider that the difference is attributed to the topological difference of them. Type-C, D and E3 are corresponding respectively to the topological states with one surface point defect, two surface and one central point defects, and one surface coiled defects, as summarized in the above-mentioned reconstructed Fig. 1. However, it is also true that it is still not easy to ascertain this from the present experimental data, because there are so many factors dominating the HIR behaviour such as directions or sizes of droplets. To make this point clearer, we added sentences mentioning this as well as the above-mentioned reconstruction of Fig. 1.

Page 14, Line 3

“On the basis of our experimental observation and analysis, the structural properties and the physical features of HIR in the observed 5 types of Ch droplets are summarized in Fig. 1. Type-A and B have the radial director configurations observed in typical N and Ch droplets [17,22,23], and hence are identical from the viewpoint of topology. As of Type-C and D, although both types have twist of defects along the cell normal, they are not topologically homeomorphic. The former has one surface point defect of +1 topological charge, while the latter has one central point defect of -1 topological charge and two surface point defects of +1 topological charge. Type-E is also a unique topological state having a surface coiled defect surrounding the droplet, supposed to be a similar state predicted by the numerical calculation in ref. 20.”

Page 16, Line 1

“In summary, the behaviour of HIR, particularly η , depends not only on the LC material and direction of the helical axis, but also the topology of the system. For example, Type-C, D, and E3 have the helical axis parallel to the direction of the applied temperature gradient. However, their absolute values of η , chirality and droplet-size dependence are quite different from each other as shown in the experimental data (Figs. 5b and 7c). These differences are obviously related to the structural and topological features, i.e. one central point defect, one central- and two surface point defects, and a surface coiled defect in type-C, D, and E3 droplets, respectively. Although we still need extensive experimental and analytical studies, such a connection between HIR and topology is very interesting and must be a key to understand further the mechanism.”

5. Page 2, line 3: HIR of what?

=> It is HIR of the director field. We modified the phrase as follows.

(Before)

Page 2, Line 3

“... heat-flux induced rotation (HIR) in a liquid crystal ...”

(After)

Page 2, Line 3

“... heat-flux induced rotation (HIR) of the director field in a liquid crystal ...”

6. Page 2, line 23: “more efficient than in the bulk” needs explanation and better description.

=> Thank you very much for the suggestion. We revised the text with additional sentences to make it more specific as follows.

(Before)

Page 2, Line 23

“... the conversion from the heat-flux to the rotation appears to be much more efficient than in bulk.”

(After)

Page 3, Line 1

“... the conversion from the heat-flux to the rotation appears to be much more efficient than in bulk, i.e. the angular velocity of HIR becomes larger in droplets than in bulk even under the same heat flux conditions. There would be two ways of reasoning in this experimental fact: (i) the director field is mobile by sliding along at the boundary because of the existence of the isotropic slippery interface in the dispersion system [7–9], (ii) the physical mechanism of HIR in the Ch droplets is intrinsically different from that in bulk [5,6,10,11].”

(New References)

[7] G. P. Bryan-Brown, E. L. Wood and I. C. Sage, *Nature*, 1999, **399**, 338.

[8] F. Nemoto, I. Nishiyama, Y. Takanishi and J. Yamamoto, *Soft Matter*, 2012, **8**, 11526.

[9] P. Oswald, *Europhys. Lett.*, 2014, **107**, 26003.

7. Page 3, lines 15-18. The authors should state more clearly that they performed experiments in other solvents and no rotation was observed.

=> For the detail of our observation on the droplet formation and their HIR behaviour in dispersions using other solvents, we modified some sentences to describe this and added four supplementary videos (new Video 1a–d.mov), including the data from a newly conducted experiment using an immiscible perfluoropolymer, CYTOP. Accordingly the texts and references are modified as follows.

(Before) Page 3, Line 16

“... However, according to our preliminary studies, no HIR could be observed in such dispersed states, even though the appearances and director structures of the dispersed Ch droplets are almost identical to those formed in the Iso-Ch transient states.

(After) Page 3, Line 19

“... dimethylpolysiloxane (DMPS) [14-18]. However, according to our preliminary studies, no HIR could be observed in such dispersed states.”

(Before) Page 5, Line 9

“Before describing the HIR dynamics of these topological Ch droplets in dispersion, we briefly examine the topologies and the rotational behaviours in the conventional transient-type non-dispersed Ch droplets formed during the Iso-Ch transition of the same LC material. mostly two types of droplets called “Striped” and “CC-type” emerge during the Iso-Ch transition as confirmed by POM (Figs. 1h and 1i) [7].”

(After) Page 5, Line 15

“Before describing the HIR dynamics of these topological Ch droplets dispersed in the

fluorinated oligomer, we briefly examined the behaviours of the same LC material in dispersions in four typical immiscible solvents, purified water, glycerol, DMPS, and poly(perfluoro-4-vinyloxy-3-methyl-1-butene) (CYTOP, AGC Chemicals). All of these dispersions showed the typical structures of Ch droplets confined by spherical boundaries with planar (water and glycerol) or homeotropic (DMPS and CYTOP) anchoring conditions [18–23]. However, none of them exhibited HIR as far as observed under POM (Video 1a-d in Supplementary Materials). Next, we checked the conventional-type HIR behaviour in the transiently formed droplets in the Iso-Ch coexisting region during the phase transition in the same LC material. In this case, mostly two types of droplets called “Striped” and “CC-type” emerge during the Iso-Ch transition (Figs. 2a and 2i), consistently to the previous study [10].”

(New References)

[15] S. Candau, P. Le Roy, and F. Debeauvais, *Mol. Cryst. Liq. Cryst.*, 1973, **23**, 283.

[16] G. E. Volovik and O.D. Lavrentovich, *Sov. Phys. JETP*, 1983, **58**, 1159.

[17] R. OndrisCrawford, E. P. Boyko, B. G. Wagner, J. H. Erdmann, S. Žumer, and J. W. Doane, *J. Appl. Phys.*, 1991, **69**, 6380.

[18] F. Xu and P. P. Crooker, *Phys. Rev. E*, 1997, **56**, 6853.

8. Page 4 line 9: early work on LC droplets is not cited, such as Volovik and Lavrentovich, and Crooker et al.

=> Thank you very much for pointing out this. We added the references to the works by Volovik and Lavrentovich as new Ref. 16 and by Crooker as new Ref. 18.

(New References)

[16] G. E. Volovik and O.D. Lavrentovich, *Sov. Phys. JETP*, 1983, **58**, 1159.

[18] F. Xu and P. P. Crooker, *Phys. Rev. E*, 1997, **56**, 6853.

9. Page 4 lines 10 -20: these structures were reconstructed recently by Posnjak et al. Using new method of director reconstruction from FCPM data, references 15 and 16. This should be stated in the revised manuscript.

=> According to your suggestion, we revised the text mentioning the FCPM (CFM) analysis by Posnjak et al. as follows.

(Before)

Page 5, Line 3

“Although the stabilities of Type-A, Type-C, and Type-D droplets show a similar tendency to that reported in the previous studies, our structural analysis gave different topological arrangements from those suggested by Orlova et al. as discussed below.”

(After)

Page 5, Line 5

“In fact, similar states to Type-A, C and E have already been reported in the literature [20-23]. Particularly, the recent work by Posnjak et al. nicely demonstrated the detailed structural models for director fields in these types by combining optical observations

using confocal fluorescence microscopy (CFM) and a theoretical approach based on calculation of elastic energy [22,23]. Our analysis for these types reconfirms their models, but we should note that the situation is slightly different because our droplets are not in ideal spherical shapes and actually in spherical-cap shapes as described later in the structural analysis section.”

10. Page 6, lines 8-10: what is the orientation of the symmetry axis of droplets with respect to heat flow direction?

=> Sorry for the unclear description. The orientation of the symmetry/winding axis is summarized in the reconstructed Fig. 1.

11. Page 7, line 13: “chaotic motion inside”. Inside what, droplet?

=> The reviewer is correct. It is “inside the droplet”. We modified it to clarify this.

(Before)

Page 7, Line 13

“...chaotic motions inside...”

(After)

Page 8, Line 5

“...chaotic motions inside the droplet...”

12. Page 14, line 8-9: how can you see the rotation of type-C structure if the direction of the helical axis is along the heat flow direction?

=> Type-C appears only with the helical axis parallel to the heat flow because of the spherical-cap shape. The rotation is observed as shown in new Fig. 4a and new Video 3. One can clearly see the rotation of the surface point defect. If your question is “how can you see the rotation of type-C structure if the direction of the helical axis is perpendicular to the heat flow direction?”, our answer is that we have never applied the in-plane temperature gradient because of technical difficulties.

13. Optical indices should be given for all materials mentioned in Materials and methods section.

=> Thank you for pointing this out. According to your suggestion, we mentioned the refractive index and birefringence of E8 and PF656 as follows. According to this, we added a new reference, Ref. 34. In fact, our Jones matrix calculation used an experimentally obtained dispersion curve of birefringence. So this was newly described in Section 6 in Supplementary Materials.

Page 19, Line 10

“The used refractive indices of PF656, and E8 for ordinary and extraordinary lights at the sodium D-line wavelength are 1.39, and 1.53 and 1.76, respectively [34].”

To Reviewer 2

Thank you very much for your invaluable opinions and suggestions. We are very much encouraged by your positive comments to our manuscript. Based on the feedbacks from the reviewers and the editor, we made a major revision on the manuscript. There are so many revisions and corrections in the text, as well as modifications of figures and Supplementary Materials. Besides, we address the answers to the questions by the reviewers one by one as follows. We believe the current version of the manuscript may be satisfactory to you. Your positive decision would be greatly appreciated.

1. The peculiarity of this work is to address the relationship between the three-dimensional chiral director field and the thermally induced rotational effect, that is found to be robust in the used arrangement. Technically, this has been possible owing to the use of a specific host fluid allowing the bulk director field to “slide” inside the spherical interface between the cholesteric and the surrounding fluid.

=> Thank you very much for the comment. We agree with you that the sliding property around the interface between the LC. Meanwhile, as answered to Reviewer 1, we think that the slight miscibility of the surrounding medium plays an important role in the effective director rotation in the LC droplet system. Since we noticed that the description for this was not sufficient, the words “slightly miscible” or “slight miscibility” are added in some parts. Besides, we added sentences mentioning this as follows.

(Before)

Page 2, Line 23

“... the conversion from the heat-flux to the rotation appears to be much more efficient than in bulk.”

(After)

Page 2, Line 23

“... the conversion from the heat-flux to the rotation appears to be much more efficient than in bulk, i.e. the angular velocity of HIR becomes larger in droplets than in bulk even under the same heat flux conditions. There would be two ways of reasoning in this experimental fact: (i) the director field is mobile by sliding along at the boundary because of the existence of the isotropic slippery interface in the dispersion system [7–9], (ii) the physical mechanism of HIR in the Ch droplets is intrinsically different from that in bulk [5,6,10,11].”

2. The fact that droplets nearby rotating droplets have fixed center of mass even (as one can see from videos) emphasize the absence of rigid-body rotation, a point that would deserve some comments.

=> The situation you mentioned is very likely. We also consider that the mass motions such as the rigid-body rotation or rotational mass flow is not realistic, because there is likely no frictional force between the droplets as elucidated in the videos. But still it is difficult to justify the absence of such mass rotations only from the present results in hands. For example, we cannot exclude the possibility of viscosity of the surrounding medium, i.e. if the surrounding medium is too viscous, the rotational flow diminishes outwards and hardly influences neighboring droplets. Based on such discussions, we

added the sentences as follows, as well as a new reference, Ref. 26 discussing the existence of the rotational flow in cholesteric droplets.

Page 9, Line 7

“Meanwhile, it would be also worth discussing, whether the observed HIR is the rotational mass motion (rotational flow or rigid-body rotation) or the simple director rotation without any mass transport. If we put our attention to one of two or more droplets gathering in the very vicinity of each other, their rotation speed is almost the same as that of standalone droplets of the same type of the same size. Besides, their centres of mass are fixed as seen in Videos 3-4 in Supplementary Materials. These mean, there are likely no frictional forces between them. This is possible, if the rotational mass motion is negligibly small or not existing at all [10,26]. Of course, this is just a speculation deduced from the observational facts and still needs further analysis by extensive studies both from theoretical and experimental approaches.”

(New Reference)

[26] G. Poy and P. Oswald, *Soft Matter*, 2016, **12**, 2604.

3. Also, I recognize the efforts made in constructing ansatz for the director field and I think that they will feed constructively further discussions on the topological features of confined chiral nematic systems. Still, in view of the proposed director field for the type-E droplets, it might be a useful information to discuss the statistics of appearance of sub-types E1-2-3 vs expected structural stability.

=> We appreciate your great suggestion. We made the statistics charts of appearances of the Type-E1, E2 and E3 droplets, as shown in new Fig. S1 in Supplementary Materials. This clearly shows the tendency described in the original text, i.e. “only Type-E1 is realized in smaller-size droplets, while in larger droplets, Type-E2 is stable for the small heat flow, and Type-E3 for the large heat flow”. According to this additional data, we added the sentences explaining this in the main text as follows.

Page 8, Line 8

“The stabilities of these types are also discussed based on the statistics of appearances as shown in Fig. S1 in Supplementary Materials. In small droplets with $2R= 15\text{--}20\ \mu\text{m}$, Type-E1 is dominant irrespective of the applied heat flux. However, in larger droplets with $2R > 20\ \mu\text{m}$, Type-E1 is obviously not preferred. Instead, Type-E2 becomes more dominant particularly in the low heat flux region. As mentioned above, Type-E2 is destabilized and transformed into Type-E3 through the unstable state by increasing the heat flux. This tendency is emphasized more in larger droplet sizes.”

4. Also, regarding the type-B that was not reported in previous study dealing with droplets with homeotropic boundary conditions, some discussion is missing. Indeed, this seems to happen because the system is actually different, hence there is less fundamental surprise than it appears in the text. In fact, here the droplets sit on substrate, which modifies the 3D confinement, hence the release of a twisted structure as the ratio p/R decreases.

=> We admit that this statement was overstated and not appropriate to describe the situation.

According to your comment, the corresponding sentence was removed. As you considered, we also think that the twisted structure in Type-B attributes to the boundary condition due to the spherical-cap shape of the droplet contacting the substrate surface. Actually, this is discussed in Page 10 Line 12–15.

(Before)

Page 4, Line 18

“... Note that to the best of our knowledge, Type-B and Type-E are observed specifically in the present system and have not been reported previously, while objectively similar states to Type-A, Type-C, and Type-D have already been reported in the literature [14-16].

(After)

Removed

(Before)

Page 5, Line 3

“Although the stabilities of Type-A, Type-C, and Type-D droplets show a similar tendency to that reported in the previous studies, our structural analysis gave different topological arrangements from those suggested by Orlova et al. as discussed below.”

(After)

Page 5, Line 5

“In fact, similar states to Type-A, C and E have already been reported in the literature [20-23]. Particularly, the recent work by Posnjak et al. nicely demonstrated the detailed structural models for director fields in these types by combining optical observations using confocal fluorescence microscopy (CFM) and a theoretical approach based on calculation of elastic energy [22,23]. Our analysis for these types reconfirms their models, but we should note that the situation is slightly different because our droplets are not in ideal spherical shapes and actually in spherical-cap shapes as described later in the structural analysis section.”

5. In short, strictly speaking, one cannot compare topological diversity of previous works with the present one, and this should appear clearly. This does not weaken the interest of the proposed spherical ansatz for the droplets.

Thank you very much for your suggestion. We agree with you that it is important to mention about the difference in the topological diversity in the present and previous works. We describe this as follows.

Page 14, Line 3

“On the basis of our experimental observation and analysis, the structural properties and the physical features of HIR in the observed 5 types of Ch droplets are summarized in Fig. 1. Type-A and B have the radial director configurations observed in typical N and Ch droplets [17,22,23], and hence are identical from the viewpoint of topology. As of Type-C and D, although both types have twist of defects along the cell normal, they are not topologically homeomorphic. The former has one surface point defect of +1 topological charge, while the latter has one central point defect of -1 topological charge and two surface point defects of +1 topological charge. Type-E is also a unique

topological state having a surface coiled defect surrounding the droplet, supposed to be a similar state predicted by the numerical calculation in ref. 20.”

Page 16, Line 2

“In summary, the behaviour of HIR, particularly η , depends not only on the LC material and direction of the helical axis, but also the topology of the system. For example, Type-C, D, and E3 have the helical axis parallel to the direction of the applied temperature gradient. However, their absolute values of η , chirality and droplet-size dependence are quite different from each other as shown in the experimental data (Figs. 3b and 4c). These differences are obviously related to the structural and topological features, i.e. one central point defect, one central- and two surface point defects, and a surface coiled defect in type-C, D, and E3 droplets, respectively. Although we still need extensive experimental and analytical studies, such a connection between HIR and topology is very interesting and must be a key to understand further the mechanism.”

List of Revisions

There are so many revisions and corrections in the text as well as modifications of the figures and Supplementary Materials. As a result, we made an extensive reorganization of the manuscript. Most of them are based on the reviewers' comments and suggestions. However not only these, but also small changes have been made. These are too numerous to list up, so that only the major points are summarized below. Please refer to the new version of the manuscript where all the revisions including small corrections are highlighted by yellow.

1. Page 2, Line 3 "... heat-flux induced rotation (HIR) of the director field in a liquid crystal ..."
2. Page 2, Line 23 "... the conversion from the heat-flux to the rotation appears to be much more efficient than in bulk, i.e. the angular velocity of HIR becomes larger in droplets than in bulk even under the same heat flux conditions. There would be two ways of reasoning in this experimental fact: (i) the director field is mobile by sliding along at the boundary because of the existence of the isotropic slippery interface in the dispersion system [7–9], (ii) the physical mechanism of HIR in the Ch droplets is intrinsically different from that in bulk [5,6,10,11]."
3. Page 3, Line 1 "... the conversion from the heat-flux to the rotation appears to be much more efficient than in bulk, i.e. the angular velocity of HIR becomes larger in droplets than in bulk even under the same heat flux conditions. There would be two ways of reasoning in this experimental fact: (i) the director field is mobile by sliding along at the boundary because of the existence of the isotropic slippery interface in the dispersion system [7–9], (ii) the physical mechanism of HIR in the Ch droplets is intrinsically different from that in bulk [5,6,10,11]."
4. Page 4, Line 18 The following sentence was removed. "... Note that to the best of our knowledge, Type-B and Type-E are observed specifically in the present system and have not been reported previously, while objectively similar states to Type-A, Type-C, and Type-D have already been reported in the literature [14-16]."
5. Page 5, Line 5 "In fact, similar states to Type-A, C and E have already been reported in the literature [20-23]. Particularly, the recent work by Posnjak et al. nicely demonstrated the detailed structural models for director fields in these types by combining optical observations using confocal fluorescence microscopy (CFM) and a theoretical approach based on calculation of elastic energy [22,23]. Our analysis for these types reconfirms their models, but we should note that the situation is slightly different because our droplets are not in ideal spherical shapes and actually in spherical-cap shapes as described later in the structural analysis section."
6. Page 5, Line 15 "Before describing the HIR dynamics of these topological Ch droplets dispersed in the fluorinated oligomer, we briefly examined the behaviours of the same LC material in dispersions in four typical immiscible solvents, purified water, glycerol, DMPS, and poly(perfluoro-4-vinyl-3-methyl-1-butene) (CYTOP, AGC Chemicals). All of these dispersions showed the typical structures of Ch droplets confined by spherical boundaries with planar (water and glycerol) or homeotropic (DMPS and CYTOP) anchoring conditions [18–23]. However, none of them exhibited HIR as far as observed under POM (Video 1a-d in Supplementary Materials). Next, we checked the conventional-type HIR behaviour in the transiently formed droplets in the Iso-Ch

coexisting region during the phase transition in the same LC material. In this case, mostly two types of droplets called “Striped” and “CC-type” emerge during the Iso-Ch transition (Figs. 2a and 2i), consistently to the previous study [10].”

7. Page 8, Line 5 “...chaotic motions inside the droplet...”
8. Page 8, Line 8 “The stabilities of these types are also discussed based on the statistics of appearances as shown in Fig. S1 in Supplementary Materials. In small droplets with $2R = 15\text{--}20\ \mu\text{m}$, Type-E1 is dominant irrespective of the applied heat flux. However, in larger droplets with $2R > 20\ \mu\text{m}$, Type-E1 is obviously not preferred. Instead, Type-E2 becomes more dominant particularly in the low heat flux region. As mentioned above, Type-E2 is destabilized and transformed into Type-E3 through the unstable state by increasing the heat flux. This tendency is emphasized more in larger droplet sizes.”
9. Page 8, Line 22 “From these results, it is obvious that η in the present dispersion system is much higher than in the previous reports for the droplets formed in the Iso-Ch coexisting region. One speculation deduced from this fact is that HIR in the current system is driven by a heat flux induced mass-flow of a little amount of oligomer molecules penetrated in the Ch droplets, analogous to the flow-induced rotation in Smectic-C* reported by Tabe et al.[25] This hypothesis is intuitively consistent with the above result showing no rotation in the dispersions using the immiscible solvents. However, to conclude so, still we need further verification of the intermediation of such a material flow, for example, by the nano-particle tracking technique.”
10. Page 9, Line 7 “Meanwhile, it would be also worth discussing, whether the observed HIR is the rotational mass motion (rotational flow or rigid-body rotation) or the simple director rotation without any mass transport. If we put our attention to one of two or more droplets gathering in the very vicinity of each other, their rotation speed is almost the same as that of standalone droplets of the same type of the same size. Besides, their centres of mass are fixed as seen in Videos 3-4 in Supplementary Materials. These mean, there are likely no frictional forces between them. This is possible, if the rotational mass motion is negligibly small or not existing at all [10,26]. Of course, this is just a speculation deduced from the observational facts and still needs further analysis by extensive studies both from theoretical and experimental approaches.”
11. Page 14, Line 3 “On the basis of our experimental observation and analysis, the structural properties and the physical features of HIR in the observed 5 types of Ch droplets are summarized in Fig. 1. Type-A and B have the radial director configurations observed in typical N and Ch droplets [17,22,23], and hence are identical from the viewpoint of topology. As of Type-C and D, although both types have twist of defects along the cell normal, they are not topologically homeomorphic. The former has one surface point defect of +1 topological charge, while the latter has one central point defect of -1 topological charge and two surface point defects of +1 topological charge. Type-E is also a unique topological state having a surface coiled defect surrounding the droplet, supposed to be a similar state predicted by the numerical calculation in ref. 20.”
12. Page 16, Line 1 “In summary, the behaviour of HIR, particularly η , depends not only on the LC material and direction of the helical axis, but also the topology of the system. For example, Type-C, D, and E3 have the helical axis parallel to the direction of the applied temperature gradient. However, their absolute values of η , chirality and droplet-size dependence are quite different from each other as shown in the experimental data (Figs. 5b and 7c). These differences are obviously related to the structural and topological features, i.e. one central point defect, one central- and two surface point defects, and a

surface coiled defect in type-C, D, and E3 droplets, respectively. Although we still need extensive experimental and analytical studies, such a connection between HIR and topology is very interesting and must be a key to understand further the mechanism.”

13. Page 19, Line 10 “The used refractive indices of PF656, and E8 for ordinary and extraordinary lights at the sodium D-line wavelength are 1.39, and 1.53 and 1.76, respectively [34].”
14. Figure 1 was reconstructed to include a table of the summary of structural features of droplets and its physical properties of HIR. All the CFM images and structural formulations are transferred to Supplementary Materials. According to these, an extensive reorganization by rearranging and reordering figures was made.
15. New references were added as follows. Accordingly, renumbering was made for all the references.

[7] G. P. Bryan-Brown, E. L. Wood and I. C. Sage, *Nature*, 1999, **399**, 338.

[8] F. Nemoto, I. Nishiyama, Y. Takanishi and J. Yamamoto, *Soft Matter*, 2012, **8**, 11526.

[9] P. Oswald, *Europhys. Lett.*, 2014, **107**, 26003.

[15] S. Candau, P. Le Roy, and F. Debeauvais, *Mol. Cryst. Liq. Cryst.*, 1973, **23**, 283.

[16] G. E. Volovik and O.D. Lavrentovich, *Sov. Phys. JETP*, 1983, **58**, 1159.

[17] R. OndrisCrawford, E. P. Boyko, B. G. Wagner, J. H. Erdmann, S. Žumer, and J. W. Doane, *J. Appl. Phys.*, 1991, **69**, 6380.

[18] F. Xu and P. P. Crooker, *Phys. Rev. E*, 1997, **56**, 6853.

[25] Y. Tabe and H. Yokoyama, *Nat. Mater.*, 2003, **2**, 806.

[26] G. Poy and P. Oswald, *Soft Matter*, 2016, **12**, 2604.

[32] J. Li and S -T. Wu, *J. Appl. Phys.*, 2004, **95**, 896

[33] R. C. Jones, *J. Opt. Soc. Am.*, 1941, **31**, 488.

[34] N. A. Vaz and G. P. Montgomery, *J. Appl. Phys.*, 1987, **62**, 3161.

REVIEWERS' COMMENTS:

Reviewer #1 (Remarks to the Author):

The authors have well addressed my comments and I can now recommend the paper for publication in Nature Communications.

Reviewer #2 (Remarks to the Author):

Second report on manuscript NCOMMS-17-17746

The revisions brought by the authors after the two reports lead me to renew my recommendation in favor of the publication.

Nevertheless, I would like to suggest to the authors to consider the following additional comment: since, as clearly mentioned in the paper, the droplets are not sphere because they are slightly wetting the substrate, there is the question whether the contact has a key role or not in the reported effect. This question is somehow left open and deserves at least to think about because the contact between the droplet and one of the two substrates of the cell leads to the appearance of a contact line. Such contact line, in presence of thermal gradients, may be the place of surface flow that could eventually drive the rotation of the droplet. Could one simply treat the substrates to prevent from the wetting, which would have the great merit to lead to the "pure" geometry of a sphere, hence facilitating the mutual direct strengthening with all previous works dealing with the topological diversity of chiral nematic droplets.

Point-by-Point Replies to Reviewers

Reviewer 1

The authors have well addressed my comments and I can now recommend the paper for publication in Nature Communications.

=> Thank you very much for your recommendation for publication in Nature Communications. We are also grateful to your constructive comments and feedback to our manuscript.

Reviewer 2

The revisions brought by the authors after the two reports lead me to renew my recommendation in favor of the publication.

=> Thank you very much for your recommendation for publication in Nature Communications. We are also grateful to your constructive comments and feedback to our manuscript.

Nevertheless, I would like to suggest to the authors to consider the following additional comment: since, as clearly mentioned in the paper, the droplets are not sphere because they are slightly wetting the substrate, there is the question whether the contact has a key role or not in the reported effect. This question is somehow left open and deserves at least to think about because the contact between the droplet and one of the two substrates of the cell leads to the appearance of a contact line. Such contact line, in presence of thermal gradients, may be the place of surface flow that could eventually drive the rotation of the droplet. Could one simply treat the substrates to prevent from the wetting, which would have the great merit to lead to the “pure” geometry of a sphere, hence facilitating the mutual direct strengthening with all previous works dealing with the topological diversity of chiral nematic droplets.

=> We deeply appreciate your kind and careful consideration. The rotation behaviour is more or less dominated by the directional differences of the symmetric axis as clearly observed in three Type-Es in the present study, as well as in CC and Striped in the previous studies. Therefore in the present manuscript, we discussed based on the fact that the major key effect of the spherical-cap shape is symmetry breaking defining the symmetric axis. Of course, we don't exclude the possibility of the existence of the effect of the contact lines at one of the surfaces as you point out. However, as you might also notice, it is still technically difficult to clarify the flow field in the droplet system and hence the problem is still left open. Regarding this situation, we added one statement as follows.

Page 16, Line7

“Nevertheless, it would be worth noting that it is still unclear whether or how the surface wetting at the contact with the substrate affects HIR in such spherical-cap geometries. For example, the modified flow fields at the contact line might influence the effect. Although we still need to perform extensive experimental and analytical studies, such a connection between the HIR and topology is very interesting and must be key to understanding the mechanism further.”

List of Revisions

There are many revisions and corrections in the manuscript. These revisions were made not only by the authors, but also by the commercial English editing service provided by Nature Research Editing Service, to improve the English text. Please find the new version of the manuscript where all the revisions including small corrections are visible by the 'track changes' function in Microsoft Word.

1. Based on the comment by Reviewer 2, we added one statement in Page 16, Line 7, "Nevertheless, it would be worth noting that it is still unclear whether or how the surface wetting at the contact with the substrate affects HIR in such spherical-cap geometries. For example, the modified flow fields at the contact line might influence the effect. Although we still need to perform extensive experimental and analytical studies, such a connection between the HIR and topology is very interesting and must be key to understanding the mechanism further."
2. Many revisions/corrections in all parts of the manuscript (title, abstract, main text, methods, references, figures, legends, and supporting materials) are made for correct formatting as pointed out by the editor.
3. The English language in the text was improved with the aid of Nature Research Editing Service.